# Isolation and Characterization of the Lytic *Pseudoxanthomonas kaohsiungensi* Phage PW916

**DOI:** 10.3390/v14081709

**Published:** 2022-08-02

**Authors:** Chang Wen, Chaofan Ai, Shiyun Lu, Qiue Yang, Hanpeng Liao, Shungui Zhou

**Affiliations:** 1Fujian Provincial Key Laboratory of Soil Environmental Health and Regulation, College of Resources and Environment, Fujian Agriculture and Forestry University, Fuzhou 350002, China; wenchang916@163.com (C.W.); aichaofanx@163.com (C.A.); lusyll123@163.com (S.L.); qiueyang2014@163.com (Q.Y.); sgzhou@soil.gd.cn (S.Z.); 2Guangdong Laboratory for Lingnan Modern Agriculture, Guangzhou 510642, China

**Keywords:** phage, *Pseudoxanthomonas kaohsiungensis*, antibiotic resistance, biological characteristics, genomic analysis

## Abstract

The emergence of multidrug-resistant bacterial pathogens poses a serious global health threat. While patient infections by the opportunistic human pathogen *Pseudoxanthomonas* spp. have been increasingly reported worldwide, no phage associated with this bacterial genus has yet been isolated and reported. In this study, we isolated and characterized the novel phage PW916 to subsequently be used to lyse the multidrug-resistant *Pseudoxanthomonas kaohsiungensi* which was isolated from soil samples obtained from Chongqing, China. We studied the morphological features, thermal stability, pH stability, optimal multiplicity of infection, and genomic sequence of phage PW916. Transmission electron microscopy revealed the morphology of PW916 and indicated it to belong to the *Siphoviridae* family, with the morphological characteristics of a rounded head and a long noncontractile tail. The optimal multiplicity of infection of PW916 was 0.1. Moreover, PW916 was found to be stable under a wide range of temperatures (4–60 °C), pH (4–11) as well as treatment with 1% (*v/w*) chloroform. The genome of PW916 was determined to be a circular double-stranded structure with a length of 47,760 bp, containing 64 open reading frames that encoded functional and structural proteins, while no antibiotic resistance nor virulence factor genes were detected. The genomic sequencing and phylogenetic tree analysis showed that PW916 was a novel phage belonging to the *Siphoviridae* family that was closely related to the *Stenotrophomonas* phage. This is the first study to identify a novel phage infecting the multidrug-resistant *P. kaohsiungensi* and the findings provide insight into the potential application of PW916 in future phage therapies.

## 1. Introduction

The genus *Pseudoxanthomonas* is a relatively newly characterized group of gamma-proteobacterium. Most species of this genus were first described in 2000 [1] and have been reportedly isolated from water, plant material, and highly contaminated soils [2]. Previous studies of *Pseudoxanthomonas* sp. found its presence to be linked to pollution in the environment, and hence it was not yet considered to be medically relevant to human health. However, evidence has since shown this genus to be obtainable from clinical human material and that it may act as an opportunistic human pathogen. For example, *Pseudoxanthomonas mexicana* had previously been derived from the urine of a 10-year-old boy with multiple catheterizations and occasional urinary tract infections in Germany [3]. A recent study has also involved the recovery of *Pseudoxanthomonas kaohsiungensis* from blood cultures obtained from an oil refinery worker in Taiwan and that it could cause chronic pericarditis [4]. Moreover, a recent correlational study characterizing the microbiomes of both oral cavities and tonsils in children with tonsillar hypertrophy suggested that *Pseudoxanthomonas* might be linked to tonsillar hypertrophy [5]. Twelve isolates of the *Pseudoxanthomonas* genus recovered from 10 patients located in two geographically separated Canadian provinces were referred to the Special Bacteriology unit at Canada’s National Microbiology Laboratory over 7 years, and some level of resistance was observed against the antibiotic classes of carbapenems and nitrofurantoin and the antibiotic group of cephems [6]. Furthermore, some *Pseudoxanthomonas spp*. were found to function as potentially lethal plant pathogens [7,8]. Although antibiotics are commonly used to treat bacterial infections, the increasing number of multidrug-resistant pathogens ultimately reduces their therapeutic efficacy over time.

Viruses that infect bacteria are known as bacteriophages, or phages for short. These are the most abundant biological entities known to exist on this planet, with approximately 10^31^ particles estimated to be present across the world [9]. Phages are found virtually everywhere, including in the soil, ocean, sludge, and extreme environments [10]. Owing to their ubiquity and abundance, phages are thought to be responsible for 20–40% of bacterial lysis events. Therefore, phages represent a major force in shaping the composition of the microbial communities present in environments [11]. Before the emergence of antibiotics, phage therapy was an efficient way to treat bacterial infections. The first use of phages as antimicrobial agents to treat human infections occurred in 1917 [12]. Over time, phages therapy in many countries were gradually replaced by the more efficient treatment option of antibiotics following their emergence in the 1940s [13]. However, in some countries, such as Poland, Georgia, phage therapy remains in use to this day [14]. Recently, with the increasing burden of antibiotic resistance on human healthcare, the applications of bacteriophages as a therapeutic tool for treating both human and plant pathogenic diseases have fostered more attention. For example, from 1981 to 1986, Hirszfeld, a Polish scientist, completed a series of overview articles and specific discussions of particular cases in 550 patients, with cure rates ranging from 75 to 100% for specific infection types [15], in addition, a survey in the Queen Astrid military hospital in Brussels, Belgium showed that a growing interest in phage therapy used as a complementary tool against antimicrobial resistant infections [16]. The increasing reports showed that phage therapy is more effective in treating some drug-resistant bacterial infections, such as patients infected with drug-resistant *Acinetobacter baumannii* and non-tuberculous *Mycobacteria* who have been successfully cured by phage therapy [17,18]. Phages have also been used in the prevention and treatment of bacterial infections in plant infections. For example, Wang et al. used a phage cocktail to control the soil-borne pathogen *Ralstonia solanacearum* [19]. Although viruses are the most abundant microbes, over 90% of viral populations remain uncharacterized [20]. Thus, it will be important to further identify and then exploit well-known and novel phage resources for applications in both agricultural environmental health and plant health.

To the best of our knowledge, no phages capable of infecting the opportunistic pathogen *Pseudoxanthomonas* genus have yet been reported. The objectives of this study were to isolate and characterize phages that possessed infectivity against *P. kaohsiungensis*. We then examined, in depth, the biological characteristics of PW916, including its morphological features, one-step growth curve, thermal and pH stability, and whole-genome sequence.

## 2. Materials and Methods

### 2.1. Isolation and Resistance Identification of Host Bacteria

A *Pseudoxanthomonas kaohsiungensis* strain was isolated from soil samples collected from Chongqing, China (29°56′ N, 106°40′ E), as described elsewhere. Briefly, approximately 5 g of fresh soil was added into a 100 mL triangular flask containing 45 mL of sterile water, incubated at 37 °C, and mixed at 160 rpm for 30 min. The soil extract was then streaked onto a 1.5% LB agar plate with a sterile inoculation ring and incubated at 37 °C overnight. Antimicrobial susceptibility tests against the strain were conducted using a previous method according to the Clinical and Laboratory Standards Institute (CLSI) guidelines 2018 [21]. The selected antibiotics were tetracycline (2 μg/mL), chloramphenicol (16 μg/mL), ampicillin (100 μg/mL), kanamycin (100 μg/mL), amoxicillin (64 μg/mL), streptomycin sulfate (30 μg/mL), gentamicin (10 μg/mL), erythromycin (15 μg/mL), and rifampicin (100 μg/mL) [22]. We determined the minimal inhibitory concentration (MIC) of Pseudoxanthomonas kaohsiungensi according to the broth dilution method [23]. The 16S rRNA gene was amplified to identify the strain by PCR using the universal primers F27 (5′-AGAGTTTGATCATGGCTCAG-3′) and R1492 (5′-TACGGTTACCTTGTTACGACTT-3′) at Sangon Biotech (Shanghai, China). The strain belongs to the genus *Pseudoxanthomonas* and was most closely related to *Pseudoxanthomonas kaohsiungensi* J36 (16S rRNA gene sequence similarity, 99.08%). The obtained *P. kaohsiungensis* was stored in the China Center for Type Culture Collection (CCTCC M 20211168). The medium used in this study was Luria–Bertani (LB) broth.

### 2.2. Phage Isolation and Purification

Phage isolation and purification were conducted using the double-agar overlay method as previously described [19]. Briefly, 10 g of fresh soil sample was suspended in 90 mL of sterile distilled water, and then centrifuged at 12,000× *g* for 2 min to remove solid particles before filtration through a 0.22 μm membrane filter to obtain the viral suspensions. A volume of 1 mL of the viral suspension and 1 mL of the bacterial suspension (10^8^ CFU/mL) were mixed into 100 mL of 1.5% LB agar and then spread onto an LB agar plate. The resulting plaques were observed following overnight incubation at 37 °C. A single clear plaque on the host bacterial lawn was selected and combined with the host at the mid-exponential phase (OD_600_ = 0.5) and cultured at 37 °C for 10–12 h or until a clear lysate was formed. The above-isolated steps were repeated five times for phage purification. The phage lysate was purified by centrifugation (12,000 rpm, 3 min, 4 °C) and filtered through a 0.22 μm filter membrane to obtain the phage suspension. Next, 10 μL of the phage dilution was dropped onto the host bacterial lawn to determine the phage titer. The phage titer was determined by spotting tenfold serially diluted phages on the bacterial lawn and this was calculated as plaque-forming units (PFU)/mL.

### 2.3. Transmission Electron Microscopy

The morphological features of phage PW916 were observed by transmission electron microscopy (TEM; model HT7700, Hitachi, Japan) with an accelerating voltage of 80 kV. In brief, 100 μL of phage suspension (10^8^ PFU/mL) was added to the lawn of host bacteria and then incubated overnight. Next, 100 μL sterile water was added to the plaque for 1 h, and then collected the rinse solution. A total of 20 μL of the phage lysate was dropped onto a 200-mesh carbon-coated copper grid for 5 min and then negatively stained with 1% (*w/v*) phosphotungstic acid for 10 min.

### 2.4. One-Step Growth Curve of the Phage

A one-step growth assay was performed according to a previously published method [24]. In brief, 50 μL of host bacteria (OD600 = 0.5, 10^8^ CFU/mL) and 50 μL of the phage suspension (10^5^–10^9^ PFU/mL) were combined in the phage-host ratios of 0.001, 0.01, 0.1, 1, and 10 in 5 mL of LB broth to culture at 37 °C for 12 h, then measured for the optimal multiplicity of infection (OMOI). Under OMOI conditions, 200 μL of the phage (10^7^ PFU/mL) and 200 μL host (10^8^ CFU/mL) suspensions were combined with 20 mL of LB broth and incubated for 5 min (37 °C, 160 rpm) to allow the phage to adsorb the bacterial cell. Next, the mixture was centrifuged at 12,000× *g* for 3 min at 4 °C to remove non-absorbed phages. Thereafter, the pellet was resuspended in 20 mL of fresh LB broth, and the phage titers in the culture were determined at intervals of 5–10 min. The phage titers at different time points were expressed in PFU/mL and plotted to generate a one-step growth curve to determine the latent period and burst size of the phage. The burst size was determined by calculating the number of phages produced divided by infected bacteria.

### 2.5. Physicochemical Stability of Phage

To test the temperature stability of the phage, 1 mL of the phage suspension (10^9^ PFU/mL) was incubated in different temperature ranges (4 to 70 °C) for 2 h. To test the pH stability of the phage, 1 mL of the phage suspension (10^8^ PFU/mL) was incubated at pH 3–12 at 37 °C for 2 h. To test the ultraviolet stability of the phage, 1 mL of the phage suspension (10^8^ PFU/mL) was incubated for 15, 30, 45, 60, and 75 min under an ultraviolet lamp (100 W). To test the chloroform tolerance of the phage, 1 mL of the phage suspension (10^9^ PFU/mL) was incubated with 1% (*w/v*) chloroform at 37 °C for 2 h. For all of these experiments, phage titers and activities were determined by plaque assay, and each experiment was repeated three independent times.

### 2.6. Phage DNA Extraction and Genome Sequencing

Phage DNA was extracted using the λ Phage Genomic DNA Extraction Kit (Abigen Biotechnology Co., Ltd., Beijing, China). DNA samples were sent to sequence (Guangdong MAGIGENE Biotechnology Co., Ltd., Guangdong, China) using the Illumina HiSeq NovaSeq 6000 platform. The raw data were filtered to obtain clean data using Soapnuke (v2.0.5) [25] and then assembled de novo using Megahit (v1.1.2) [26]. The assembled results were compared with the clean reads using BWA (v0.7.17), and this was followed by the calculation of the read utilization and GC content of the genome. The prodigal was used to predict the putative open reading frames (ORFs) with default parameters by -p mode of single [27]. Finally, the protein sequences of ORFs were annotated by comparing the viral sequences against the UniProtKB/Swiss-Prot Database using the BLASTP algorithm [28] with the best hit of e-value < 10^−3^. The phylogenetic trees based on the major capsid protein (*mcp*), terminase large subunit (*terL*), and DNA polymerase alpha subunit (*pola1*) genes were constructed using MEGA (Version 7.0.18) (the neighbor-joining algorithm; 1000 bootstrap replications).

### 2.7. Statistical Analysis

Statistical analysis for significance was performed using the SPSS v20.0 (SPSS Inc., Chicago, IL, USA). The differences between treatments were analyzed using the Student’s *t*-tests where *p* values below 0.05 were considered statistically significant. A circle map of the genome was generated using the CGview Tool (http://cgview.ca/, accessed on 15 December 2021) [27]. To generate the arrow diagram of gene function annotations, ggplot2 and gggenes in the R Package were used. Mauve [29] and Easyfig [30] were used for collinearity analysis with homologous phage whole-genome sequences from NCBI and IMG/VR databases. To clarify the taxonomy of the phage, phylogenetic trees were constructed based on genetically significant sequences, using the adjacent method of MEGA Software (v6.0) (bootstrap set to 1000) [31]. Finally, the whole-genome sequence and related annotated information were uploaded to GenBank (https://www.ncbi.nlm.nih.gov/, accessed on 20 December 2021) under the accession number (OL960029).

## 3. Results and Discussion

### 3.1. Phage Isolation, Morphology, and Characteristics

The potential opportunistic pathogen *P. kaohsiungensis* was isolated from soil and subsequently found to possess multidrug resistance to kanamycin, amoxicillin, streptomycin, gentamicin, and erythromycin, suggesting it to be a multidrug-resistant strain (Appendix A) [22]. The colony grown on LB were circular, and the middle of the colony is elevated darker yellow, the edge is smooth light yellow (Appendix A). Phage PW916 was isolated using *P. kaohsiungensis* as a host from soil collected from Chongqing, China. The phage formed a clear circular plaque with a diameter of approximately 1 mm on the host bacterial lawn (Figure 1a). Some phage particles were found to be accumulated and absorbed onto the surface of the *P. kaohsiungensis* cells (black arrows), while others were observed within the *P. kaohsiungensis* cells (white arrows) (Figure 1b), thus confirming the in situ assembly of progeny phages in the host [32]. Additionally, PW916 was observed to possess a head with a diameter of 55 ± 3.0 nm and a long noncontractile tail with a length of 150 ± 3.0 nm, while no tail fibers were observed (Figure 1c). Consequently, based on its basic morphology, PW916 was proposed to belong to the *Caudoviricetes* class according to the International Committee on Taxonomy of Viruses (ICTV) [33]. To the best of our knowledge, PW916 is the first reported phage of *P. kaohsiungensis*. The morphology of PW916 was similar to that of previously reported phages that infect *Xylella*, *Xanthomonas*, and *Stenotrophomonas* [8,34]. Therefore, PW916 may be harnessed as an agent for phage therapy in the treatment of *P. kaohsiungensis* infections.

### 3.2. Multiplicity of Infection and One-Step Growth Curve of PW916

The optimal multiplicity of infection (OMOI) of PW916 was 0.1; briefly, phages at 10 times lower than the bacterial concentration are better (Figure 2a). A one-step growth curve was plotted under the conditions of the OMOI with the initial adsorption rate of 93.33% (Figure 1b). The results revealed that PW916 was a lytic phage because it had a short latency period of 5 min (there was no significant change in the phage titer within 0–5 min), then followed by a rise period of 75 min, and a growth plateau of 40 min. The burst size of PW916 was approximately 203 virions per cell. We compared these characteristics to other published lytic phages, such as *Caudoviricetes* class phage vB_SmaS_BUCT548, which was reported to exhibit an incubation period of 30 min and a burst size of 134 PFU/cell [35,36]. Based on its characteristics, PW916 appears to be a good candidate for treating drug-resistant *P. kaohsiungensis* infections due to its lytic ability.

### 3.3. Thermal, pH, Chloroform, and UV Stability of Phage PW916

We investigated the stability of phage PW916 under different conditions, such as temperature, pH, UV, and organic solvents. The thermal stability of PW916 was tested from 4 to 70 °C for 2 h. The results showed no significant reduction in phage stability following incubation temperatures between 4 and 50 °C while exhibiting a sharp decrease at temperatures above 50 °C and being completely inactivated at 70 °C (Figure 3a). The results of the pH stability experiment revealed that PW916 exhibited broad pH stability ranging from pH 5–11, while PW916 had a higher tolerance to alkaline conditions compared to acidic conditions. However, PW916 was completely inactivated at both pH 3 and pH 12 (Figure 3b), indicating that strongly acidic and alkaline conditions could inhibit the activity of the phage. The pH stability of PW916 was superior to that of the *Acinetobacter* phage which only exhibited stability at pH 4–8 [24]. The phage titers of PW916 significantly (*p* < 0.05) decreased from 10^8^ PFU/mL to 10^5^ PFU/mL after 15 min of UV exposure and completely inactivated after 90 min of the treatment (Figure 3c). This result was similar to that obtained from other phages, likely due to structural proteins associated with the adsorption capacity in phage particles having been destroyed by the UV exposure [37]. The infectivity of phage PW916 was stable, as the phage titer showed no clear change following treatment with 1% (*v/w*) chloroform for 2 h at 37 °C (*p* > 0.05), thus indicating that PW916 was not sensitive to chloroform (Figure 3d). This may be due to the fact that phage PW916 was determined not to contain capsid lipids which are typically sensitive to chloroform [38]. Taken together, these results indicate that phage PW916 retains high activity under a wide range of physical and chemical conditions, hinting at its potential application in the future.

### 3.4. Whole-Genome Analysis of Phage PW916

The genome of phage PW916 has a GC content of 62.63% and a length of 47,760 bp (Figure 4a). Sixty-four ORFs were identified in the PW916 genome. The longest ORF was 2903 bp and the shortest was 335 bp, and these encoded the central tail hub and DnaJ domain protein, respectively (Table 1). No tRNA genes, antibiotic resistance genes, or virulence factors were detected in the PW916 genome, suggesting that this phage poses a low risk of mediating the horizontal gene transfer of antibiotic resistance [39]. The PW916 functional gene annotations were divided into five modules, namely DNA replication, DNA packaging, auxiliary metabolism, phage structure, and lytic proteins. A total of 39 other ORFs were identified to code hypothetical proteins (Figure 4b). The DNA replication module was determined to include DNA polymerase I, DNA ligase, DNA helicases, primase, and RecB exonucleases. In the presence of ATP, the RecB exonucleases function as multifunctional nucleases in the phage genome [40], suggesting that they participate in the synthesis of genomic raw materials for the initiation of recombination and the repair of defective recombination outcomes [41]. Both ORF36 and ORF37 corresponded to the terminase large subunit and portal protein, which form the ATP-hydrolyzing motor for DNA packaging [42]. The terminase large subunit possesses endonuclease and ATPase activities, and it can cleave DNA strands and provide energy for the unidirectional movement of DNA strands [43]. The portal protein is a circular homolog with a central channel oligomer that provides a pathway for DNA injection [44]. Structural proteins of the virus were annotated from 22,951 to 40,831 bp of the genome. The capsid assembly proteins included the head morphogenesis protein, scaffold protein, major capsid protein, structural protein, and head–tail joining protein (ORF38-ORF44). The tail assembly module was organized starting with the gene encoding the major tail protein (ORF45) as well as the tape measure protein, tail assembly protein, and central tail hub protein. This module followed the typical gene architecture for *Siphoviridae* morphogenesis [45].

Additionally, PW916 encoded auxiliary metabolism enzymes related to the thymidylate synthesis pathway. ORF62 is a nucleoside triphosphate pyrophosphohydrolase that can hydrolyze nucleoside triphosphate to produce its corresponding nucleoside monophosphate [46]. Putative deoxycytidylate (dCMP) deaminase (ORF58) processes dCMP to produce deoxyuridine monophosphate (dUMP) for thymidylate synthase. Other enzymes that participate in this pathway include deoxyuridylate hydroxymethyltransferase and methyltransferase. With regard to the lytic protein, ORF54 has been found to encode an endolysin that participates in the final stage of the lytic cycle and plays an indispensable role in the release of mature phage particles [47]. As such, it was considered a potential candidate for use against antibiotic resistance due to its lytic activity [48]. These results indicate that PW916 may participate in host-assisted metabolic processes and potentially impact the metabolic functioning of the host.

### 3.5. Comparative Genomics and Phylogenetic Analysis of Phage PW916

For a comparative genomic analysis of the phage, the whole genomic sequence of PW916 was searched against the virus databases NCBI and IMG/VR by BLAST. Seven similar phages were detected from the NCBI Database (Table 2), including a phage infecting *Stenotrophomonas*, two phages infecting *Xylella*, three phages infecting *Caulobacter,* and another phage belonging to *Caudovirales*. Additionally, 15 similar virus genome contigs (genome completeness >85%) were mapped from the IMG/VR Database (Appendix A). However, only four virus contigs were predicted to be associated with known hosts, and all of these hosts belonged to *Proteobacteria*, including *Thioalkalivibrio* and *Sinorhizobium*. No similar viral genome was mapped to infect the same host of the *Pseudoxanthomonas* genus in the virus database. The highest sequence similarity was assigned to the *Stenotrophomonas* phage vB_SmaS-AXL_3 (MT536174.2), with 89.64% identity and 89% coverage. According to gene-coding proteins and morphological comparisons, PW916 and vB_SmaS-AXL_3 both belonged to *Caudoviricetes* class and possessed high similarity in both structural proteins and auxiliary factors [34]. However, there were clear differences between PW916 and vB_SmaS-AXL_3 in their gene arrangement structures (Appendix A).

To investigate the taxonomic position of PW916, phylogenetic trees were constructed using MEGA, using the three conserved proteins of the terminase large subunit, DNA polymerase I, and major capsid protein, respectively (Figure 5). By comparing these protein sequences against the NCBI virus database, PW916 was found to be closely related to phages that infect *Stenotrophomonas*, *Pseudomonas*, *Enterobacter*, and *Salmonella*. The phylogenetic trees based on the three genes revealed that PW916 was closest to the *Stenotrophomonas* phage which also belonged to the *Caudoviricetes* class. The bacteria *Stenotrophomonas* and *Pseudoxanthomonas* share a close relationship as they belong to the same family of Xanthomonadaceae [36]. Taken collectively and combined with the results of the phylogenetic tree and the comparative genomic analysis, the highest identity of phage PW916 and the *Stenotrophomonas* phage was less than 90%, indicating that phage PW916 was considered a new virus species that can infect *P. kaohsiungensis*.

## 4. Conclusions

This study presents a characterisation of a newly isolated phage PW916. To the best of our knowledge this is the first isolated phage able to infect the multi-drug re-sistant host *P. kaohsiungensis*. PW916 exhibited lytic activity and relatively high thermotolerance and acid tolerance, thereby showing great potential in the control of *P. kaohsiungensis* infection across a variety of conditions. However, further studies are required to assess the potential of PW916 to control *P. kaohsiungensis* infections as no antibiotic resistance gene and virulence gene of ORFs were identified in the genome of this phage, promoting the potential application for phage therapy in the future. Because of the absence of virulent genes and an antibiotic resistance gene, the phage in our study was safe for use as phage therapy for the prevention and control of *P. kaohsiungensis*. Our study has identified a promising candidate for phage therapy as well as established a foundation for further studies on the interaction of hosts and their phages.

## Figures and Tables

**Figure 1 viruses-14-01709-f001:**
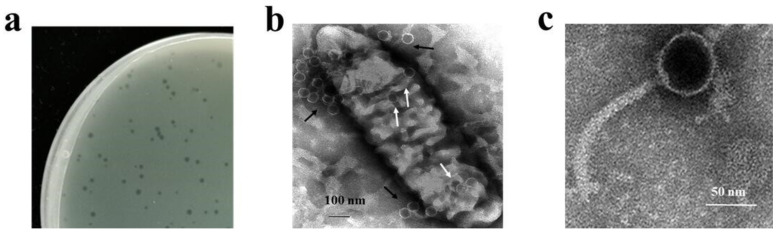
Morphology of plaques and phage PW916. (**a**) Plaque morphology. Numerous phage particles infecting the host (**b**) and virion morphology of phage PW916 (**c**) were observed by TEM.

**Figure 2 viruses-14-01709-f002:**
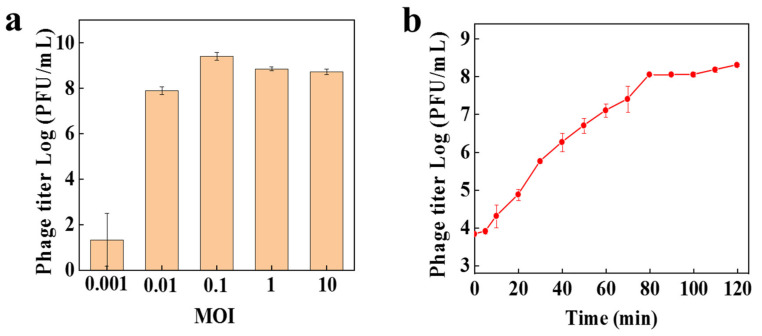
Physiological characteristics of phage PW916. (**a**) Multiplicity of infection (MOI) of phage PW916. (**b**) One-step growth curve of phage PW916.

**Figure 3 viruses-14-01709-f003:**
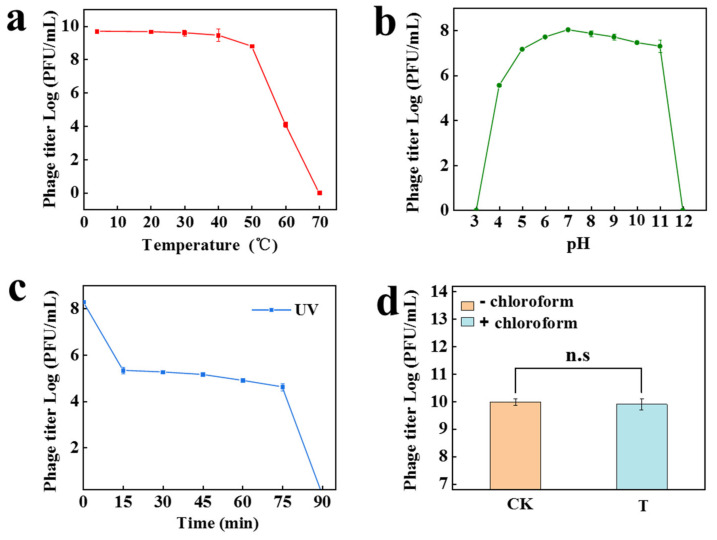
Biophysical characterization of phage PW916 under varied conditions. Thermostability (**a**), pH stability (**b**), UV tolerance (**c**), and chloroform stability (**d**) of phage PW916. Data are displayed as the means plus standard deviations (SD) calculated from three independent experiments and ‘n.s’ denotes for nonsignificant difference. In (**d**), CK: no chloroform, T: with chloroform.

**Figure 4 viruses-14-01709-f004:**
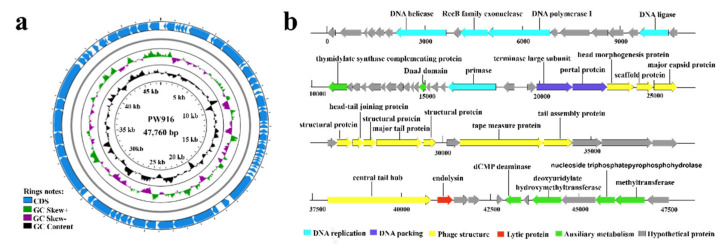
Genomic analysis of phage PW916. (**a**) Genome map of phage PW916. Concentric rings denote the following features (from the inner to outer rings): nucleotide positions are forward strand (outer); GC skew is (G − C)/(G  +  C) (gray: −; purple:  +). The predicted genes located on the genome are labeled on the outer rings. (**b**) Structure of the putative phage genes based on the predicted reading frames of phage PW916.

**Figure 5 viruses-14-01709-f005:**
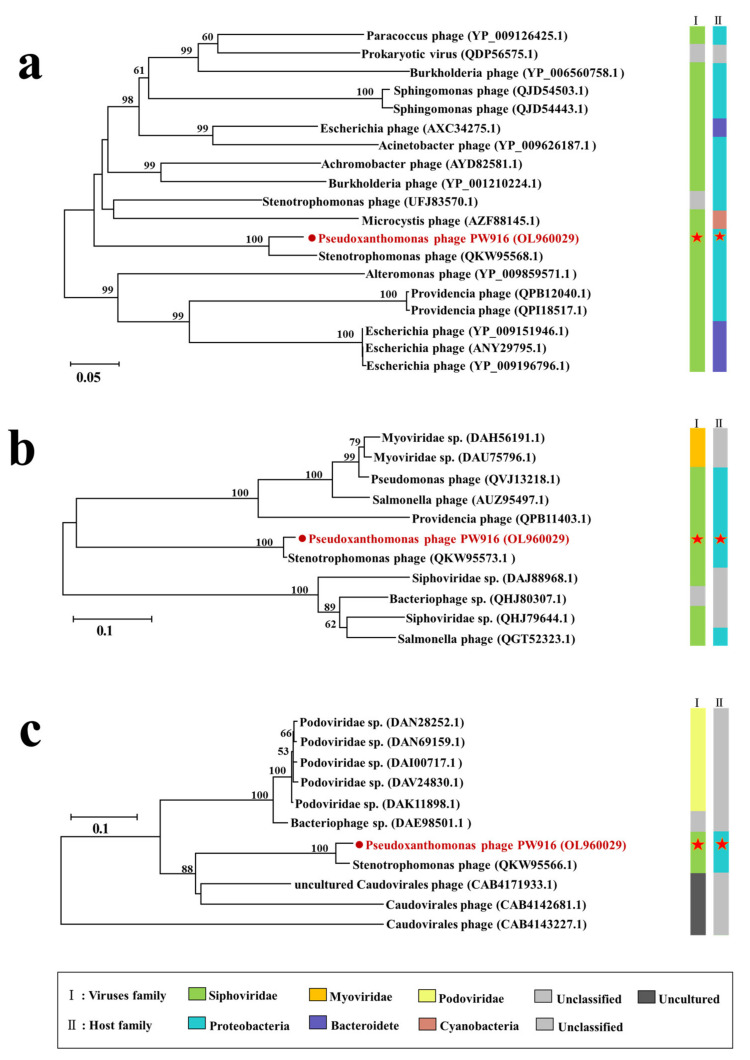
Phylogenetic relationships of phage PW916. (**a**) Phylogenetic tree of the terminase large subunit. (**b**) Phylogenetic tree of the major capsid protein. (**c**) Phylogenetic tree of the DNA polymerase alpha subunit. The tree was generated by ClustalW alignment of amino acid sequences using the neighbor-joining method and a bootstrap value of 1000 iterations.

**Table 1 viruses-14-01709-t001:** Functional annotations of open reading frames (ORFs).

ORF	Start	Stop	Length (bp)	Strand	Start Codon	Function
7	3646	2093	1553	-	ATG	DNA helicase
9	4919	4068	851	-	ATG	RecB family exonuclease
10	6838	4919	1919	-	ATG	DNA polymerase I
16	10,479	9574	905	-	ATG	DNA ligase
18	11,591	10,743	848	-	ATG	Thymidylate synthase complementing protein
29	15,047	14,712	335	-	ATG	DnaJ domain
33	18,098	15,984	2114	-	ATG	Primase
36	19,871	21,430	1559	+	ATG	Terminase large subunit
37	21,442	22,947	1505	+	ATG	Portal protein
38	22,951	24,132	1181	+	ATG	Head morphogenesis protein
39	24,238	24,972	734	+	ATG	Scaffold protein
40	25,006	26,022	1016	+	ATG	Major capsid protein
42	26,410	26,934	524	+	ATG	Structural protein
43	26,939	27,322	383	+	ATG	Head–tail joining protein
44	27,319	27,750	431	+	ATG	Structural protein
45	27,755	29,308	1553	+	ATG	Major tail protein
46	29,337	29,768	431	+	ATG	Structural protein
48	30,566	33,406	2840	+	ATG	Tape measure protein
49	33,417	34,376	959	+	ATG	Tail assembly protein
53	37,928	40,831	2903	+	ATG	Central tail hub
54	40,998	41,447	449	+	ATG	Endolysin
58	43,361	42,915	446	-	ATG	Deoxycytidylate deaminase
60	44,481	43,663	818	-	GTG	Deoxyuridylate hydroxymethyltransferase
62	45,992	45,426	566	-	ATG	Nucleoside triphosphate pyrophosphohydrolase
63	46,827	46,003	824	-	ATG	Methyltransferase

**Table 2 viruses-14-01709-t002:** Comparative genomic analysis of phage PW916 with the NCBI Database.

Accession No.	Length (bp)	Coverage	Identity	E-Value	Scientific Name
MT536174.2	47,545	89%	89.64%	0.0	Stenotrophomonas phage vB_SmaS-AXL_3
NC_042345.1	55,601	1%	78.44%	7 × 10^−77^	Xylella phage Salvo
NC_052973.1	56,232	0	78.79%	3 × 10^−75^	Xylella phage Bacata
KY555144.1	218,729	0	86.96%	2 × 10^−07^	Caulobacter phage Ccr5
KY555143.1	220,299	0	86.96%	2 × 10^−07^	Caulobacter phage Ccr2
KY555142.1	219,348	0	86.96%	2 × 10^−07^	Caulobacter phage Ccr10
MK527152.1	486	0	84.42%	2 × 10^−07^	Caudovirales sp. GX_16_bay_2_59859
AP014685.1	9,780,023	0	76.27%	4 × 10^−10^	Bradyrhizobium diazoefficiens NK6
AP023108.1	9,278,204	0	76.27%	4 × 10^−10^	Bradyrhizobium diazoefficiens XF19
AP023105.1	9,278,205	0	76.27%	4 × 10^−10^	Bradyrhizobium diazoefficiens XF16

## Data Availability

Not applicable.

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
