# Peer review of "Isolation and Characterization of the Lytic Pseudoxanthomonas kaohsiungensi Phage PW916"

_viruses, 2022, doi:10.3390/v14081709_

Round 1

Reviewer 1 Report

A paper is relatively well-written and the methodology is adequately explained (there is some room for improvement though). However, the authors did not mention what was the Pseudoxanthomonas taxonomy before such genus has been created as the bacteria itself certainly existed. The authors states several times that this is the first report describing phages against Pseudoxanthomonas. Maybe it is not but it was described in the literature under different name. This issue is crucial and must be investigated and clarified. For instance many Raoultella strains (and phages against them) 20 years ago were identified as Klebsiella. Only taxonomy has changed, not phages and their bacterial hosts.

Line 56: 1031 particles - lack of superscript and this is global issue throughout the manuscript - please correct in all instances.

Line 62: clinical trials from 1919? Do authors understand the nature of clinical trials? I don't think we can talk about "clinical trials" that took place in 1919.

Line 63: "Over time phages were gradually replaced". This statement is only partially true. There are countries where phage therapy has not been replaced completely (Poland, Georgia). Please correct.

Line 69: This citation (13) is difficult to understand. First of all, the cited authors treated only one patient - not enough to draw any conclusions regarding phage treatment. Secondly, even the title of this paper describe "adjunctive treatment" which was applied together with antibiotics. We don't really know what caused an improvement of the patient condition - phages, antibiotics or both? I don't think this paper is appropriate to support phage treatment. There are several highly-cited papers describing successful phage treatment in dozens or hundreds of patients - please cite them. 

Line 97: The 16S rRNA was amplified to identify the strain. I'd like to read something more about bacteria identification (no less important than phage identification). Was it confirmed using different platforms? What about colony morphology and cultivation proces in the lab? This is also related to my first comment - what kind of bacteria is that?

Line 115: Please define "high-titer".

Line 177: The bacterial host was isolated from soil, not from the patient. It is not clinical strain only environmental thus all statements regarding "promising candidate for phage therapy" are exaggerated. Unless the authors refer only to phage therapy in agriculture.

Fig. 1a This image is not acceptable. It is way too small, plaques are not clearly visible - please change it. In fact, Fig. 1b and 1c should be of larger size as well.

Line 207. How many bacterial strains (including clinical ones) have been tested against this phage to support statement on “strong lytic ability”? What is the lytic spectrum of this phage?

Fig. 3d – Explain “n.s”

Lines 327-333. Section with conclusions is too short. A real plague in the current literature focusing on phages are never-ending phage therapy connotations made by the authors who have no experience in phage therapy at all. Furthermore, such optimistic statements are supported by the outcomes derived from one or two patients treated with antibiotics. I will not accept any paper with such bold therapeutic statements not supported by strong clinical evidence. 

Author Response

We thank the reviewer’s contributions to our manuscript.

Reviewer 2 Report

Thank you for the opportunity to review this manuscript. 

The article is intresting, well written and structured. However there are few changes required in order to corectly characterise P. kaohsiungensis:

- Please defined multidrug-resistance according to the definition proposed by Magiorakos et al. (2012) and the natural resistance. More so in the cited article (row 43) it is not specify that the bacterial strain is MDR.

-In Table S1 please remove the antibiotics that P. kaohsiungensis is naturaly resistant to, according to CLISI/EUCAST.

In Introduction section, row 56, please confirm that the number 1031particles is correct according to the reference cited.

Author Response

We thank the reviewers for their professional advice, which greatly improved our manuscript.

Reviewer 3 Report

The research article by Wen et al. on 'Isolation and characterisation of the novel phage PW916 for lysing the Multidrug-resistant Pseudoxanthomonas Kaohsiungensi' is an interesting study that reports a new member of the phage community that infects P. kaohsiungensi. Though the article reports a new phage within this community, the content is the same for any phage researcher with no new identified genes/proteins in its genome. 

Minor:

1. The manuscript needs thorough English correction/typos and bacterial names should be italicised.

2. Methods: TEM methodology should be explained. What concentration of phages were used? How did the authors perform bacteria-phage interaction using TEM?

3. Methods and Results: What is the adsorption rate of the phage? The short latent period may be due to the unpredicted adsorption rate/time. In figure 2b, explain the latent period and burst size.

4. Methods: Why de novo assembly was performed? what was the reference seq used?

5. It is better to explain the MOI in the case of phage against bacteria ratio. For example, an MOI of 0.1 is better which means phages at 100 times lower than the bacterial concentration are better.

6. Figure 3D, what are CK and T in the x-axis?

7. Figure 5 can be removed. It is not relevant to this study's results. 

8. Line no. 328: Write as 'To the best of our knowledge...'

Author Response

(The authors gave the same response as above.)
